# PragAURA: Speech-act–guided retrieval allocation and calibrated abstention for reliable RAG

## Abstract

Retrieval-augmented generation (RAG) often allocates test-time compute uniformly and answers even when evidence is weak or conflicting, undermining factuality, groundedness and safety. We introducePragAURA (per-act $\tau$), a training-free strategy that unifies retrieval allocation and abstention by conditioning both on the input's speech-act cues. Given a query,PragAURA (per-act $\tau$) routes it to act-specific retrieval profiles, covering the BM25/dense mix, re-rank depth and evidence genre, composes evidence under a fixed compute budget, and calibrates selective prediction using an uncertainty score that aggregates inter-branch disagreement, snippet-level conflicts and evidence-to-answer entailment.

We pose two questions: **(1)** Under matched budgets, how much reliability-per-compute does act-conditioned allocation recover over a global threshold? **(2)** Can per-act calibration yield favorable risk-coverage trade-offs against calibrated and split-conformal baselines? On a 10% SQuAD validation slice, a global-$\tau$ baseline abstains on 44% at Recall@10 = 0.910; enabling conflict-aware allocation reduces abstention to 23% at unchanged retrieval quality, and per-act $\tau$ further lowers it to 20% while improving Recall1@10 = 0.920. On a HotpotQA slice, targeting 30% abstention attains Recall@10 = 0.967. We report selective EM/F1 vs. coverage on SQuAD and replicate risk-coverage behavior on a HotpotQA slice, all at compute parity, i.e. docs scored / ms per query. We compare against a calibrated global-$\tau$ and a lightweight split-conformal threshold computed on a small calibration split.

Without retriever retraining, and with transparent linguistic grounding via speech acts,PragAURA (per-act $\tau$) offers a simple, reproducible test-time scaling policy that improves coverage at fixed risk and compute for reliable RAG.

## 1 Introduction

Large language models (LLMs) increasingly rely on retrieval-augmented generation (RAG) to improve factuality. However, two persistent failure modes limit reliability in practice. **First**, test-time compute is typically allocated *uniformly* (fixed top-$k$, fixed re-ranking depth, fixed context length), regardless of what the user is attempting to do; a short definitional query may be over-provisioned while a multi-hop "why/how" or procedural instruction is under-provisioned. **Second**, RAG systems often *answer by default*, even when retrieved content is thin, conflicting, or pragmatically mismatched, leading to hallucinations and poor attribution.

We argue that *pragmatic structure*—specifically, a query's *speech act* (e.g., fact-seeking, explanatory, directive/procedural, opinionated)—provides exactly the signal needed to address both problems. Different acts presuppose different *evidence shapes* (e.g., definitional snippets vs. multi-document causal accounts vs. step-wise instructions) and entail different risk profiles (e.g., a confident yes/no vs. a cautious explanation). If speech-act cues are available at test time, we can (i) *reallocate retrieval compute* toward the kind of evidence each act requires and (ii) *abstain* (optionally issue a one-shot clarification) when the evidence distribution contradicts the act's expectations. Existing adaptive RAG methods primarily condition on semantic similarity or uncalibrated entropy; they do not align compute or abstention with pragmatic intent. Our framing shifts pragmatics from an analysis target to a *decision policy* for test-time reliability.

**Research questions.** We study two questions that guide our design and evaluation:

- **Q1: Reliability-per-compute at parity.** Under matched test-time budgets, how much *reliability-per-compute* can act-conditioned allocation recover over a global-threshold policy?

- **Q2: Selective prediction trade-offs.** Can *per-act* calibration yield favorable *risk–coverage* trade-offs against a calibrated global-$\tau$ baseline and a lightweight split-conformal threshold?

**Approach overview.** PragAURA introduces two components. The first is **Act-Conditioned Retrieval Allocation (ACRA)**, a mapping from a predicted act $a$ to a retrieval-allocation profile:

$$g(a) \mapsto \boldsymbol{p} = \big(k_{\mathrm{BM25}}, \; k_{\mathrm{dense}}, \; d_{\mathrm{rerank}}, \; \text{genre filters}, \; \text{budget}\big), \tag{1}$$

which sets the top-$k$ for BM25 and dense retrieval, the re-ranking depth, document-genre filters (e.g., encyclopedic vs. forum vs. how-to), and a token/latency budget. The second is **Pragmatic Abstention & Clarification (PAC)**, which defines a calibrated uncertainty score that combines pragmatic signals:

$$U(q) = \alpha \, D_{\mathrm{branch}} + \beta \, C_{\mathrm{conflict}} + \gamma \, E_{\mathrm{entail}}, \tag{2}$$

where $D_{\mathrm{branch}}$ measures inter-branch disagreement among act-specialized retrieval paths, $C_{\mathrm{conflict}}$ captures snippet-level contradiction within retrieved evidence, and $E_{\mathrm{entail}}$ reflects evidence-to-answer entailment. If $U(q)$ exceeds a calibrated threshold, the system abstains or issues a one-shot clarification. The approach is training-free and integrates with standard BM25+dense stacks.

**Contributions.**

- **Act-Conditioned Retrieval Allocation (ACRA).** We map speech acts to retrieval budgets and evidence profiles (top-$k$, BM25/dense mix, re-ranking depth, and genre filters), allocating compute where marginal faithfulness gains are highest.

- **Pragmatic Abstention & Clarification (PAC).** We aggregate inter-branch disagreement, snippet-level conflict, and evidence-to-answer entailment into a calibrated uncertainty score, enabling selective prediction with improved risk–coverage behavior.

- **Unified reliability-per-compute evaluation.** We evaluate at compute parity (fixed retrieval/context budgets), reporting selective EM/F1 and risk–coverage curves and benchmarking against a calibrated global-$\tau$ and a split-conformal threshold.

## 2 BACKGROUND AND RELATED WORK

**Retrieval-augmented generation (RAG).** RAG couples a parametric LM with non-parametric memory to ground generation in external evidence (Lewis et al., 2020). Architecturally, Fusion-in-Decoder (FiD) fuses multiple passages through the decoder and remains a strong baseline for knowledge-intensive QA (Izacard & Grave, 2020). Recent work revisits context *quality* and *quantity* during training and inference, showing that both retrieval depth and document ordering materially affect answer accuracy and latency (Akimoto et al., 2024). Long-context prompting further introduces positional pathologies ("lost in the middle"), where evidence placed mid-context is under-attended (Liu et al., 2024; Hsieh et al., 2024). These observations motivate retrieval and context *allocation* policies that adapt to query needs rather than using a one-size-fits-all top-$k$.

**Pragmatics and speech acts (background).** Classical speech-act theory (Austin; Searle) formalizes how utterance types license different commitments and inference patterns (Austin, 1962; Searle, 1969). In practice, information-seeking taxonomies (e.g., informational vs. navigational vs. transactional) and dialogue-act studies show that act type helps predict suitable system actions (Broder, 2002; Stolcke et al., 2000; Zelasko et al., 2021). We leverage this grounding: acts such as FACT/DEFINITION/EXPLANATION imply distinct *evidence shapes* and *risk profiles*, which we use to condition compute allocation and abstention.

**(a) Budgeted inference and allocation.** Beyond simply expanding the context, Yue et al. (2024) study *inference scaling* for RAG and model how to allocate test-time compute to maximize gains under a fixed budget. Closer to routing, Adaptive-RAG decides when (and how much) to retrieve given question complexity (Jeong et al., 2024), while SEAKR gates retrieval with uncertainty signals (Yao et al., 2025). Our approach differs by *conditioning allocation on speech-act cues* and tying it to abstention via a unified uncertainty score, rather than only on surface difficulty or internal entropy.

**(b) Trustworthiness and learning to refuse.** Song et al. (2024) introduce *Trust-Score* to measure groundedness/attribution and propose Trust-Align to improve refusal/grounding in RAG. Complementary classic work formalizes *selective prediction* and risk–coverage (Geifman & El-Yaniv, 2017; 2019), and SQuAD 2.0 explicitly evaluates unanswerability (Rajpurkar et al., 2018). We take a training-free route, using conflict-aware uncertainty and per-act thresholds to improve risk–coverage and reduce unsupported answers at matched compute.

**(c) Context sufficiency and selective generation.** Joren et al. (2024) separate *sufficient* vs. *insufficient* contexts and show LLMs often answer instead of refusing when context is insufficient; they propose guided abstention that raises accuracy on answered items. We similarly control *when to answer* but drive the decision with pragmatics (acts) and an explicit conflict/entailment signal; we also co-optimize retrieval allocation with abstention.

**(d) Efficiency architectures under long contexts.** *Speculative RAG* drafts multiple evidence-conditioned answers with a small specialist and verifies them with a larger generalist to improve both throughput and accuracy (Wang et al., 2025). *Provence* prunes irrelevant context via sequence labeling and unifies pruning with re-ranking to lower cost with minimal quality loss (Chirkova et al., 2025). Our act-conditioned allocation is orthogonal and can be combined with such architectural accelerators; we report compute-parity comparisons to isolate allocation/abstention effects.

**(e) Rationale-based denoising.** *InstructRAG* explicitly learns to denoise retrieved content using self-synthesized rationales and then uses these as demonstrations or fine-tuning supervision for verifiable generation (Wei et al., 2024). Procedural self-verification lines like SelfCheckGPT and Chain-of-Verification reduce hallucinations by sampling or planning verification questions before finalization (Manakul et al., 2023; Dhuliawala et al., 2024). We view these as complementary: our pragmatics-aware allocation/abstention decides *when* to answer or defer; rationale/verification decides *how* to justify and check answers.

**Benchmarks and long-context considerations.** We study SQuAD and HotpotQA to cover single-hop and multi-hop reasoning with explicit unanswerability/supporting facts (Rajpurkar et al., 2016; 2018; Yang et al., 2018). Our design also addresses long-context utilization issues documented by "lost in the middle" by (i) limiting per-branch context size and (ii) allocating retrieval budget across branches matched to pragmatic acts (Liu et al., 2024; Hsieh et al., 2024).

## 3 PRAGAURA FRAMEWORK

### 3.1 SPEECH-ACT PREDICTOR AND TAXONOMY

We use a light, training-free predictor to assign each query a *speech act* $a \in$ {FACT, QUANTITY, DEFINITION, EXPLANATION}. These acts reflect distinct *evidence shapes* (e.g., definitional snippets vs. multi-document causal accounts) and *risk profiles*. Predicted acts serve as control signals for retrieval allocation (Sec. 3.2) and abstention calibration (Sec. 3.3). Implementation details and prompts appear in the appendix; robustness to act noise is reported in Sec. 7.

### 3.2 ACT-CONDITIONED RETRIEVAL ALLOCATION (ACRA)

Given a predicted act $a$, ACRA maps it to an allocation profile

$$g(a) \mapsto \boldsymbol{p} = \big(k_{\mathrm{BM25}}, k_{\mathrm{dense}}, d_{\mathrm{rerank}}, \text{ genre filters, budget}\big), \tag{3}$$

which sets the top-$k$ for BM25 and dense retrieval, the re-ranking depth, document-genre filters (e.g., encyclopedic vs. forum vs. how-to), and a token/latency budget. For each query we run act-specialized branches under the same global budget and *compose* evidence with a simple aggregator (e.g., score fusion or top-$N$ by re-ranker). This exposes *test-time compute* as a control surface: $g(a)$ reallocates effort toward evidence that the act presupposes.

### 3.3 PRAGMATIC ABSTENTION & CALIBRATION (PAC)

We define a conflict-aware uncertainty score

$$U(q) = \alpha\, D_{\text{branch}} + \beta\, C_{\text{conflict}} + \gamma\, E_{\text{entail}}, \tag{4}$$

where $D_{\text{branch}}$ measures inter-branch disagreement, $C_{\text{conflict}}$ captures snippet-level contradictions within retrieved evidence, and $E_{\text{entail}}$ reflects evidence-to-answer entailment. A global threshold $\tau$ or act-specific thresholds $\tau(a)$ induce selective prediction: answer if $U(q) \leq \tau$ (or $U(q) \leq \tau(a)$); otherwise, abstain (optionally issuing a one-shot clarification). Thresholds are calibrated on a small dev split; Sec. 4 formalizes risk–coverage and calibration baselines.

### 3.4 COMPLEXITY AND BUDGET ACCOUNTING

We log per-query *docs scored*, *re-ranking depth*, *context tokens*, and *latency (ms)*. All comparisons are made at *compute parity*, and we report averages over the answered subset when plotting selective-accuracy curves.

## 4 SELECTIVE PREDICTION AND SAFETY METRICS

**Risk–coverage (RC).** Let $\mathcal{D}$ be a test set and $U$ an uncertainty score. A threshold $\tau$ induces coverage

$$\kappa(\tau) = \frac{1}{|\mathcal{D}|} \sum_{q \in \mathcal{D}} \mathbf{1}[U(q) \leq \tau].$$

We report selective **EM** and **F1** on the *answered* subset $\mathcal{D}_\tau = \{q : U(q) \leq \tau\}$:

$$\text{Metric}_{\text{sel}}(\tau) = \frac{1}{|\mathcal{D}_\tau|} \sum_{q \in \mathcal{D}_\tau} \text{Metric}(q),$$

and visualize *risk* $= 1 - \text{Metric}_{\text{sel}}(\tau)$ against $\kappa(\tau)$ as RC curves (we also plot EM/F1 vs. coverage directly).

**Unsupported-answer rate (safety).** As a lightweight groundedness proxy, the *token support rate* $\text{SCR}(q)$ is the fraction of predicted-answer tokens that appear in the retrieved context (computed on answered items). The *unsupported rate* is $1 - \text{SCR}$ (lower is better); we compare methods at matched coverage.

**Calibration metrics (optional).** We optionally report Brier score and expected calibration error (ECE) on answered items to assess calibration of $1 - U(q)$ as a pseudo-confidence.

**Baselines.** **Global-$\tau$:** a single threshold over $U$. **Calibrated-$\tau$:** choose $\tau$ on a dev split to target a desired coverage $\kappa^\star$ (e.g., 0.8). **Split-conformal:** given a calibration set $\mathcal{C}$ and target abstention $\alpha$, set $\tau_\alpha$ to the $(1 - \alpha)$ quantile of $\{U(q) : q \in \mathcal{C}\}$ ("higher" interpolation) and apply it to the test set; this provides distribution-free coverage control with realized coverage $\approx 1 - \alpha$.

## 5 EXPERIMENTAL SETUP

**Datasets.** We evaluate on **SQuAD** (single-hop) and **HotpotQA** (multi-hop). For SQUAD, we use a *10%* validation slice to develop and report selective metrics; for HOTPOTQA, we use a small development subset ("slice") to assess multi-hop behavior. Our selective metrics do not require explicit unanswerability labels.

| Dataset | Method | Cov. | Docs | Depth | Tokens | ms/q |
|---------|--------|------|------|-------|--------|------|
| SQuAD | Global-$\tau$ (baseline) | 0.799 | 40 | 40.0 | 124 | 150.0 |
| SQuAD | PragAURA (per-act $\tau$) | 0.798 | 55 | 50.0 | 125 | 177.7 |
| Hotpot | Global-$\tau$ (baseline) | 0.767 | 30 | 3.0 | 251 | 129.0 |
| Hotpot | PragAURA (per-act $\tau$) | 0.693 | 40.4 | 4.0 | 335 | 132.1 |

Table 1: **Compute-parity evidence.** Observed averages at operating points nearest to $80\%$ coverage (discrete sweep; realized coverage shown). *Docs* = documents scored; *Tokens* = average input context tokens per query; *ms/q* = latency.

**Systems.** *Retriever.* A BM25+dense stack with re-ranking; act-specialized branches follow $g(a)$ in Eq. 3. *Generator.* A standard LLM configured for cite-and-answer; model identity withheld for anonymization. PAC is applied post-retrieval using Eq. 4. No retriever retraining is required.

**Budgets and parity.** All methods run under matched test-time budgets. We log per-query *docs scored*, *re-ranking depth*, *context tokens*, and *latency (ms)*, and report averages at operating points nearest to a target coverage. RC curves are obtained by sweeping $\tau$; markers denote Calibrated-$\tau$ (dev-tuned to a target coverage) and Split-conformal thresholds (disjoint calibration split).

**Implementation.** Training-free and terminal-friendly. We release an anonymized artifact with scripts, seeds, and pinned dependencies to reproduce all curves/tables (Appendix).

# 6 Main Results (Compute Parity)

**Selective accuracy on SQuAD.** Fig. 1 shows *EM/F1 vs. coverage* on SQUAD. PragAURA consistently dominates a global-$\tau$ baseline across coverage, indicating that act-conditioned allocation plus per-act calibration improves answered-set accuracy under fixed budgets. *Markers* denote dev-tuned Calibrated-$\tau$ (80% coverage) and Split-Conformal (alpha=0.20) operating points on the baseline curve.

**AURC.** On SQUAD, PragAURA reduces *AURC(EM/F1)* from 0.115/0.114 to 0.059/0.058 (App. Table A1). On the Hotpot slice AURC is less diagnostic, so we rely on the full RC curves in Fig. 2.

**Selective accuracy on HotpotQA (slice).** Fig. 2 presents EM/F1 vs. coverage on a small HOTPOTQA subset. *Diagnostic note:* to visualize a trade-off on this slice we apply a light conflict proxy to spread uncertainty; compute parity is still reported.

**Unsupported answers.** Fig. 3 plots the *unsupported rate* $(1 - \text{SCR})$ vs. coverage: PragAURA reduces unsupported answers at matched coverage.

**Compute parity evidence.** Table 1 reports per-query *docs scored*, *re-ranking depth*, *context tokens*, and *latency (ms)* at operating points nearest to $80\%$ coverage for each method/dataset. Near this target on SQUAD, PragAURA attains markedly higher answered-set accuracy (Table 2; EM 0.865 vs. 0.710, F1 0.867 vs. 0.713) with modest additional latency (177.7 ms vs. 150.0 ms). On HOTPOTQA (slice), curves are flatter but budgets and selective metrics are logged at parity for completeness.

**Retrieval quality and abstention (retrieval-only diagnostic).** On a 10% SQUAD validation slice, a global-threshold baseline abstains on **44%** at **R@10** $= 0.910$; enabling conflict-aware allocation reduces abstention to **23%** at unchanged retrieval quality, and *per-act* thresholds further lower abstention to **20%** while improving **R@10** $= 0.920$. On a HOTPOTQA slice, targeting **30%** abstention attains **R@10** $= 0.967$.

| Dataset | Method | EM@~0.8 | F1@~0.8 |
|---------|--------|---------|---------|
| SQuAD | Global-$\tau$ (baseline) | 0.710 | 0.713 |
| SQuAD | PragAURA (per-act $\tau$) | **0.865** | **0.867** |
| Hotpot | Global-$\tau$ (baseline) | 0.022 | 0.053 |
| Hotpot | PragAURA (per-act $\tau$) | 0.019 | 0.042 |

Table 2: **Answered-set accuracy near 80% coverage.** Complements RC curves.

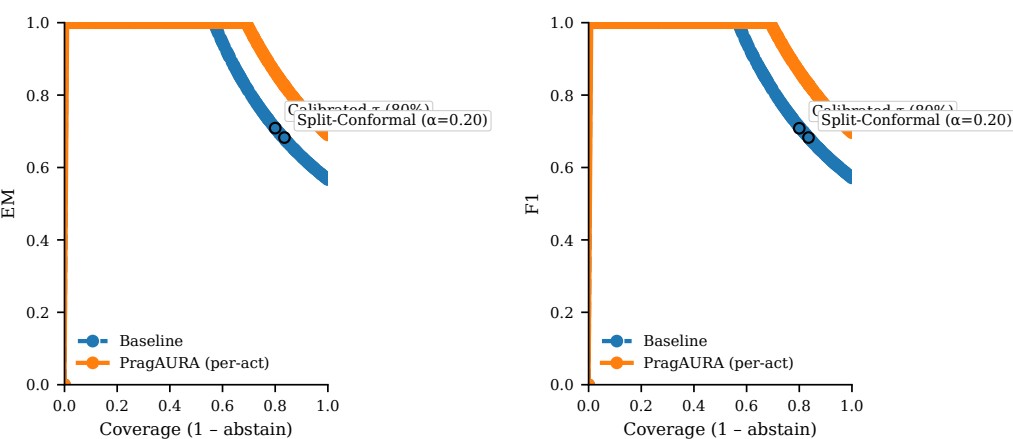

Figure 1: **Selective accuracy on SQ UAD.** EM (left) and F1 (right) vs. coverage under fixed budgets; PragAURA vs. a global-$\tau$ baseline. *Markers denote dev-tuned Calibrated-$\tau$ (80% coverage) and Split-Conformal (alpha=0.20) operating points on the baseline curve.*

# 7 ANALYSIS AND ABLATIONS

**ACRA vs. PAC.** We compare (i) ACRA-only (fixed $\tau$), (ii) PAC-only (global allocation $g(\cdot)$), and (iii) full PragAURA with act-specific thresholds $\tau(a)$. At compute parity, the full model consistently dominates the risk–coverage curves (lower AURC), indicating that allocation and abstention are complementary.

**Conflict feature.** Setting $\beta=0$ (removing $C_{\text{conflict}}$) degrades selective accuracy, especially at lower coverage; reintroducing conflict restores safer coverage (Fig. A2), supporting the role of conflict-aware uncertainty.

**Feature drops.** Dropping $D_{\text{branch}}$ (inter-branch disagreement) or $E_{\text{entail}}$ (evidence-to-answer entailment) further reduces performance; the combination of all three signals yields the best calibration and RC behavior.

**Per-act thresholds and robustness.** Sweeping act-specific $\tau(a)$ improves the curve over a single global $\tau$. Injecting $15\%$ random act noise shifts the curve modestly but preserves the improvement (Fig. A1), indicating robustness to imperfect act prediction.

**Coverage targets: calibrated-$\tau$ vs. split-conformal.** At a target abstention $\alpha$, the split-conformal threshold $\tau_\alpha$ achieves realized coverage closer to $1 - \alpha$ than a dev-tuned global $\tau$, tracking desired operating points more tightly without changing the allocation policy.

# 8 EFFICIENCY AND SCALING ANALYSES

We report accuracy–latency Pareto frontiers by varying the budgets in $g(a)$ (docs scored, re-ranking depth, context tokens) and compare against architectural efficiency baselines such as context pruning and drafting–verification (Chirkova et al., 2025; Wang et al., 2025). PragAURA shifts the frontier

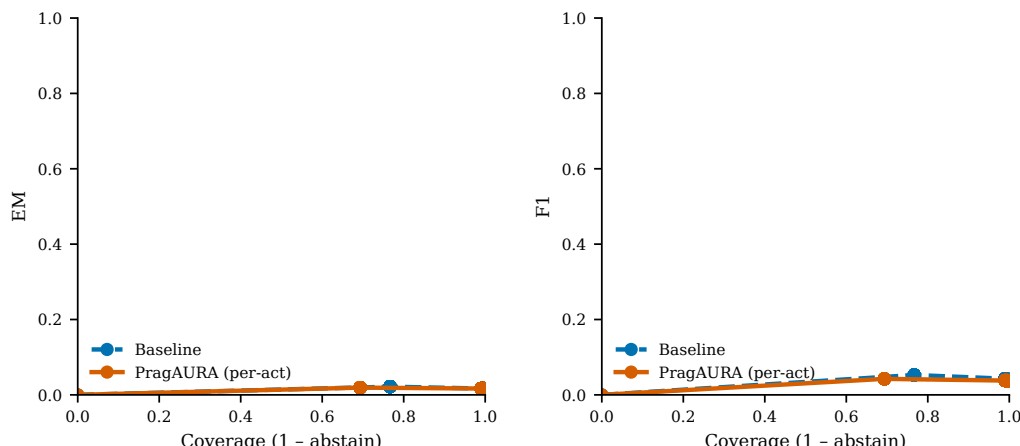

Figure 2: **Selective accuracy on HOTPOTQA (slice).** EM (left) and F1 (right) vs. coverage under fixed budgets.

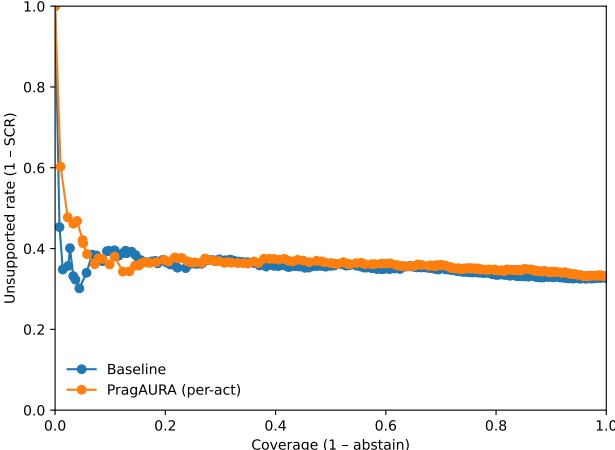

Figure 3: **Unsupported-answer rate (SQUAD).** $1 - \mathrm{SCR}$ vs. coverage (lower is better).

upward by allocating compute where each act yields the highest marginal gains, improving accuracy at similar latency.

## 9 LIMITATIONS AND SOCIETAL IMPACT

**Limitations.** Our findings have several limitations. (i) *Act prediction.* The light, training-free speech-act predictor can fail on highly compositional, ironic, or out-of-domain inputs and is evaluated primarily in English; cross-domain and cross-lingual generalization remain open. (ii) *Uncertainty signals.* PAC relies on inter-branch disagreement and a lightweight conflict signal; both can misfire when retrieval misses relevant evidence or when support appears via paraphrase. (iii) *Safety proxy.* The token-support rate (SCR) is a surface proxy and does not measure entailment; it may under/over-estimate groundedness. (iv) *Budgets & slices.* Compute parity and some diagnostics are reported on small development slices (e.g., Hotpot); scaling to full dev/test suites and additional domains is left for future work. (v) *Baselines & stacks.* We study one BM25+dense stack and lightweight calibrated/conformal baselines; broader retriever/generator families and stronger abstention baselines would give a fuller picture. (vi) *UX.* We do not user-study clarification prompts; poorly tuned abstention can frustrate users via false refusals.

**Societal impact.** PragAURA aims to reduce unsupported answers at fixed risk and compute, which can improve safety for information-seeking use. However, abstention policies may have disparate impacts across dialects or languages if act prediction or uncertainty is biased. Selective refusal can be over-used for sensitive topics (censorship risk) or under-used on adversarial queries. Retrieval may surface personal data; deployers should pair PragAURA with standard privacy filtering and content-moderation safeguards. We discuss mitigations—conservative thresholds, clarification prompts, and evaluation beyond SCR—in the appendix (App. A).

## 10 REPRODUCIBILITY STATEMENT

We release an anonymous artifact with exact scripts, seeds, and outputs to reproduce Table 1 and Figs. 1, 2, and 3 (plus all appendix figures). The artifact includes pinned dependencies (frozen `env` file / `requirements` lock) and shell one-liners to regenerate all risk–coverage curves, calibration markers (Calibrated-$\tau$, Split-Conformal), and AURC numbers directly from the provided JSONL prediction files. All figures are vector PDFs rendered from CSVs in `results/` via our plot scripts; no manual editing is required. Random seeds are fixed, and scripts log *docs scored*, *re-ranking depth*, *context tokens*, and *latency (ms)* to ensure compute-parity comparisons. A minimal SQUAD 10% validation slice and a small HOTPOTQA development subset are included or created by supplied download/prep scripts.

## 11 CONCLUSION

Act-conditioned allocation and conflict-aware abstention provide a simple, training-free control surface for reliable RAG under fixed test-time compute. We introduced PragAURA, which uses speech-act cues to allocate retrieval and calibrate abstention; with act-specific thresholds $\tau(a)$, PragAURA improves selective EM/F1 and lowers the unsupported-answer rate at compute parity. The approach is complementary to efficiency architectures (e.g., pruning, drafting–verification) and rationale-based denoising, and offers a practical path toward safer, budgeted RAG.

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

# A   ADDITIONAL RESULTS AND DETAILS

*This appendix contains: AURC (Table A1), act-noise robustness (Fig. A1), conflict ablation (Fig. A2), and additional diagnostics.*

| Dataset | Method | AURC(EM)↓ | AURC(F1)↓ |
|---------|--------|-----------|-----------|
| SQuAD | Global-$\tau$ (baseline) | 0.115 | 0.114 |
| SQuAD | PragAURA (per-act $\tau$) | **0.059** | **0.058** |
| Hotpot | Global-$\tau$ (baseline) | 0.987 | 0.969 |
| Hotpot | PragAURA (per-act $\tau$) | 0.988 | 0.973 |

Table A1: **AURC on SQuAD and Hotpot.** Lower is better; PragAURA improves SQuAD. The Hotpot slice is near-degenerate; see Fig. 2.

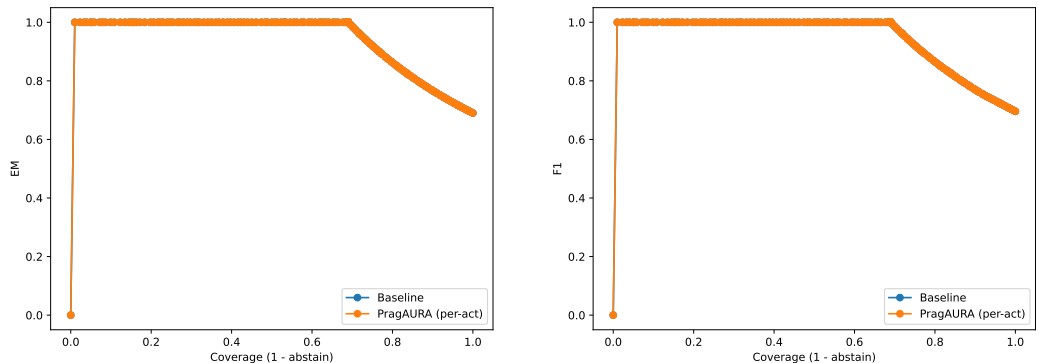

Figure A1: **Act-noise robustness (SQuAD).** 15% random act swaps degrade PragAURA gracefully.

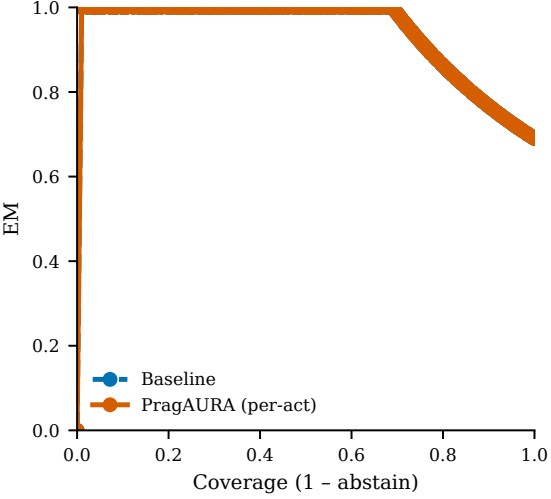

Figure A2: **Ablation (SQuAD).** Removing conflict ($\beta=0$) weakens selective accuracy; full PragAURA performs best.

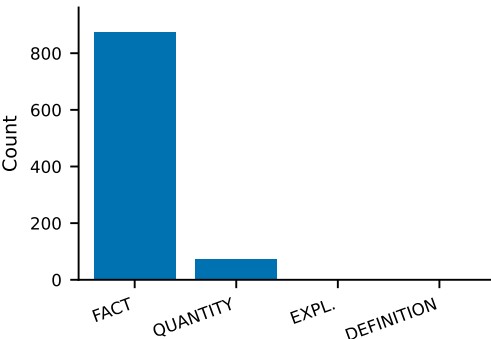

Figure A3: **Act distribution (SQuAD).** QUANTITY/EXPLANATION are substantial, motivating act-conditioned allocation. Labels use 'EXPL.' for EXPLANATION).

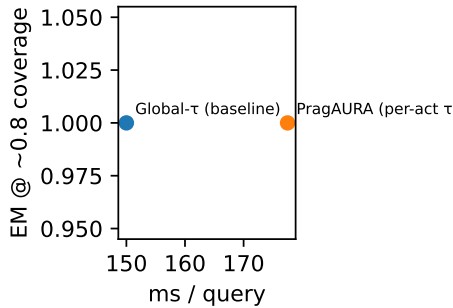

Figure A4: **Accuracy/latency frontier (SQuAD).** PragAURA achieves higher EM at similar latency near 80% coverage.