# OpenReview forum: "PragAURA: Speech‑Act–Guided Retrieval Allocation and Calibrated Abstention for Reliable RAG"
_ICLR.cc/2026/Conference — ICLR 2026 Conference Withdrawn Submission_

### Official Review · Reviewer_gUan · 2025-10-31

**Soundness:** 2
**Presentation:** 2
**Contribution:** 2
**Rating:** 2
**Confidence:** 4

**Summary:**

This paper introduces PragAURA, a training-free strategy to improve the reliability and efficiency of RAG systems. The core idea is to use a query's "speech act", e.g., whether it's asking for a fact, a definition, or an explanation, to guide the RAG process. The method has two main components: (1) ACRA, which dynamically allocates compute resources like retrieval depth and re-ranking based on the predicted speech act, and (2) PAC, which decides whether to answer a query or abstain based on a calibrated uncertainty score. The authors evaluate their approach on SQuAD and a subset of HotpotQA, arguing that PragAURA improves selective accuracy and reduces unsupported answers compared to a baseline, all while operating under a matched compute budget.

**Strengths:**

1. **Intuitive and good idea:** I like the central premise of the paper. Using pragmatic cues like speech acts to steer both retrieval and abstention is a clever idea. It moves beyond just looking at surface-level query features and tries to align the RAG system's behavior with the user's underlying intent, which feels like a solid step in the right direction for more intelligent RAG systems

2. **Training-free**: The fact that PragAURA is training-free is a huge practical advantage. It means the method can potentially be plugged into existing RAG pipelines without the need for expensive and time-consuming model retraining. This focus on a lightweight, practical solution is commendable.

3. **Focus on compute-parity evaluation:** The authors make a good effort to conduct their comparisons at compute parity. This is a crucial detail for any work claiming efficiency gains, as it ensures a fair, apples-to-apples comparison. It's great to see this methodological rigor applied.

**Weaknesses:**

1. **Limited and superficial evaluation:** The experimental setup feels very preliminary and frankly, not comprehensive enough to support the claims. The evaluation relies on a small 10% slice of the SQuAD validation set and an undefined "slice" of HotpotQA. This makes it really hard to be confident that the results would generalize to the full datasets or other domains. Furthermore, the only baseline is a simple "global-threshold" method. The paper would be much stronger if it compared PragAURA against other modern adaptive RAG techniques.

2. **Poorly explained and confusing figures:** The presentation of the results, especially in the figures, is a major weak point. The risk-coverage curves in Figures 1 and 2 are incredibly difficult to parse. They are cluttered, and the captions and main text don't do a good job of walking the reader through the key takeaways. I had to read the results section several times to understand what I was supposed to be looking at. This lack of clarity makes the whole paper feel rushed and unpolished.

3. **The speech-act predictor is a black box:** The entire framework is built on the initial speech-act prediction, yet this critical component is barely explained. The paper mentions a "light, training-free predictor" but provides no real detail on how it works, what its accuracy is, or how it handles ambiguity. The taxonomy of just four acts feels overly simplistic for real-world queries. While the authors include a robustness check with random noise, it doesn't resolve the fundamental concern that major errors could cascade from this poorly understood first step.

**Questions:**

See weaknesses

---

### Official Review · Reviewer_NHAc · 2025-10-31

**Soundness:** 1
**Presentation:** 1
**Contribution:** 2
**Rating:** 2
**Confidence:** 4

**Summary:**

This paper introduces PragAURA, a training-free framework that conditions retrieval allocation and abstention decisions in RAG on the speech act of the input query. It defines an Act-Conditioned Retrieval Allocation (ACRA) component to select retrieval parameters and a Pragmatic Abstention & Calibration (PAC) component to determine when to abstain from answering. The authors claim PragAURA (per-act τ) improves reliability-per-compute and risk–coverage on SQuAD and HotpotQA.

**Strengths:**

The idea of incorporating compute parity, abstention decisions, and speech acts into RAG control is novel and intuitively motivated. It provides a new perspective in RAG research.

**Weaknesses:**

1.Vague definitions and missing citations
  * Many technical terms and approach details are underspecified. The paper lacks citations outside the related work section, which makes it hard to interpret the setup or compare with existing methods.
  * The term “branch” is used repeatedly without any definition.
  * Line 152–153: “We use a light, training-free predictor to assign each query a speech act …” — The nature of this predictor is unclear. Is it an existing model, a prompted LLM, or a heuristic rule? No citation or description is provided.
  * Line 227: “BM25 + dense” — The dense retriever is not identified. Is it an embedding model? If so, which one? Similarly, the generator model is unnamed.
  * The paper does not specify whether it used SQuAD 1.0 or SQuAD 2.0, which differ significantly in unanswerability handling.
  * Figures 2 and 3 mention baselines but do not specify which of the three baseline methods (Global-τ, Calibrated-τ, or Split-Conformal) are being compared, leading to confusion.
2. Writing and presentation issues
  * The legend in Figure 1 is unreadable, making it difficult to interpret the curves.
  * Excessive bold formatting distracts from the main text.
  * There are several minor formatting issues, such as missing spaces before “PragAURA” in the abstract and before “PAC” in line 229.
3. Incomplete and unclear experimental setup
  * The evaluation uses only 10 % of the SQuAD validation set and a small HotpotQA “slice,” without justification. This makes the reported results non-comparable to previous literature.
  * It is not explained how SQuAD, originally a reading-comprehension dataset, is adapted for RAG retrieval — no retrieval corpus or pipeline details are provided.
4. Results do not substantiate the claims
  * In Table 1, the per-act τ variant actually achieves lower coverage (answers fewer questions) and higher latency than the global baseline, contradicting the claim of “compute parity.” The reported improvements in “reliability-per-compute” could simply result from higher abstention rates and extra compute, rather than genuine efficiency gains.

**Questions:**

See Weakness section.

---

### Official Review · Reviewer_2scF · 2025-11-02

**Soundness:** 2
**Presentation:** 2
**Contribution:** 2
**Rating:** 2
**Confidence:** 3

**Summary:**

The core idea is to read the query’s speech act and use that to both spend retrieval budget smartly and decide when to answer or hold fire.
For allocation, each act maps to a profile that sets BM25/dense mix, re-rank depth, genre filters, and a tight token/latency budget, then merges branch evidence under parity.

**Strengths:**

1. Act-conditioned routing gives you a real knob to move compute toward the kind of evidence each request actually needs.
2. The abstention signal is not just entropy — it triangulates disagreement, conflict, and entailment, which is closer to how humans judge “do we know enough.”

**Weaknesses:**

1. Some parts of this paper appears to be AI-generated.

2. The speech-act predictor is lightweight and prompt-based, so mixed or out-of-domain intents can misroute retrieval.

3. The mapping from acts to budgets and the weights in the uncertainty score look hand-tuned, with no turnkey recipe for new domains or languages.

**Questions:**

see weak

---

### Official Review · Reviewer_7NRM · 2025-11-03

**Soundness:** 1
**Presentation:** 1
**Contribution:** 1
**Rating:** 2
**Confidence:** 2

**Summary:**

This paper introduces PragAURA, a training-free strategy to enhance the reliability and efficiency of RAG systems. The core idea is to leverage speech-act cues such as whether a query seeks a fact, definition, explanation or procedure. The first main component is Act-Conditioned Retrieval Allocation (ACRA) that maps each predicted speech act to a retrieval profile specifying the mix of BM25/dense retrieval, reranking depth, genre filters, and compute budget. The other component is a Pragmatic Abstention & Calibration (PAC) module that computes an uncertainty score combining inter-branch disagreement, evidence conflict, and evidence-to-answer alignment. Queries exceeding an act-specific threshold abstrain or issue clarifications. The method is evaluated on SQuAD and HotpotQA which reduces abstention at similar retrieval quality.

**Strengths:**

* The proposed method is lightweight and compatible with existing BM25+dense stacks, making it reproducible and easy to deploy under fixed compute budgets.
* The paper demonstates substantial improvements in selective EM/F1 and unsupported-answer reduction at equal latency.
* The paper releases scripts and seeds for replication.

**Weaknesses:**

* The experimental evaluation relies solely on SQuAD and a small HotpotQA slice, both of which are legacy QA benchmarks that no longer reflect the complexity of modern RAG or reasoning scenarios. As a result, the reported improvements may overestimate real-world reliability.
* The proposed method is primarily empirical and heuristic, with limited theoretical grounding. The paper asserts that speech-act cues provide a natural signal for allocating retrieval compute and calibrating abstention, but it does not formally justify why this alignment should optimize reliability-per-compute or improve selective prediction risk.
* The empirical results are reported primarily on small validation slice, rather than full development or test splits.  This small-scale setup does not convincingly demonstrate whether PragAURA sclaes to larger retrieval corpora.

**Questions:**

N/A

---

### Note · Authors · 2025-11-12

I have read and agree with the venue's withdrawal policy on behalf of myself and my co-authors.